# Qualitative Phenotyping of Obstructive Sleep Apnea and Its Clinical Usefulness for the Sleep Specialist

**DOI:** 10.3390/ijerph17062058

**Published:** 2020-03-20

**Authors:** Marcello Bosi, Andrea De Vito, Danny Eckert, Joerg Steier, Bhik Kotecha, Claudio Vicini, Venerino Poletti

**Affiliations:** 1Pulmonary Operative Unit, Department of Thoracic Diseases, Morgagni-Pierantoni Hospital, Romagna Health Company, 47121 Forlì, Italy; marcello.bosi@libero.it; 2Head & Neck Department, Ear Nose Throat (ENT) Unit, Santa Maria delle Croci Hospital, Romagna Health Company, 48121 Ravenna, Italy; 3Adelaide Institute for Sleep Health, A. Flinders University. Centre of Research Excellence, Adelaide 5049, Australia; d.eckert@neura.edu.au; 4Lane Fox Unit/Sleep Disorders Centre, Guy’s & St Thomas’ National Health Service (NHS) Foundation Trust, London SE19RT, UK; joerg.steier@gstt.nhs.uk; 5Centre of Human & Aerospace Physiological Sciences, Faculty of Life Sciences and Medicine, King’s College, London WC2R 2LS, UK; 6Nuffield Health Brentwood, Shenfield Road, Brentwood, Essex CM15 8EH, UK; bhikkot@aol.com; 7Head & Neck Department, ENT & Oral Surgery Unit, Morgagni-Pierantoni/Infermi Hospital, Romagna Health Company, 47121 Forlì, Italy; claudio@claudiovicini.com; 8Ear Nose and Throat (ENT) Clinic, Special Surgery Department, Arcispedale S. Anna Hospital, Ferrara University, 44124 Ferrara, Italy; 9Department of Otolaryngology Head and Neck surgery, S.Orsola-Malpighi University Hospital, 40138 Bologna, Italy; 10Pulmonary Operative Unit, Department of Thoracic Diseases, Morgagni-Pierantoni Hospital, Romagna Health Company, 47121 Forlì, Italy; venerino.poletti@auslromagna.it; 11Department of Respiratory Diseases & Allergy, Aarhus University Hospital, 8200 Aarhus, Denmark

**Keywords:** obstructive sleep apnea, phenotypization, CPAP, pathophysiological traits

## Abstract

Introduction: The anatomical collapsibility of the upper airway, neuromuscular tone and function, sleep–wake and ventilatory control instability, and the arousal threshold all interact and contribute to certain pathophysiologic features that characterize different types of obstructive sleep apnea (OSA). A model of qualitative phenotypizationallowsus to characterize the different pathophysiological traits in OSA patients.Methods: A narrative review was performed, to analyze the available literature evidence, with the purpose of generating a model of qualitative phenotypization to characterize pathophysiological traits in patients with OSA.Results: 96 out of 3829 abstracts were selected for full-text review. Qualitative phenotyping model of OSA:Data concerning the OSA qualitative pathophysiological traits’ measurement can be deducted by means of clinical PSG, grade of OSA severity, and therapeutic level of Continuous Positive Airway Pressure (CPAP) and are reported in the text. This approach would allow qualitative phenotyping with widely accessible methodology in a routine clinical scenario and is of particular interest for the sleep specialist, surgical treatment decision-making, and customized OSA multimodality treatment.

## 1. Introduction

Obstructive sleep apnea syndrome (OSA) is the most common type of sleep-disordered breathing (SDB), and it symptomatically affects up to 9% of men and 4% of women [1]. Excessive sleepiness [2,3], neurocognitive deficiency [4,5,6], and low quality of life [7,8] are the most frequent presenting complaints, with road traffic accidents [9] and increased cardiovascular morbidity and mortality [10,11] potentially being the most severe complications reported; OSA leads to significant increases of health and social costs [12].

The anatomical collapsibility of the upper airway (UA), neuromuscular tone and function, sleep–wake and ventilatory control instability, and the arousal threshold (AT) all interact and contribute to certain pathophysiologic features that characterize different types of OSA [13].

This narrative review analyzes available evidence in the literature, with the purpose of generating a model of qualitative phenotypization, to characterize pathophysiological traits (PTs) in patients with OSA, using clinical polysomnography (PSG), with a focus on the assessment of anatomical collapsibility of the UA.This is of particular interest for the sleep specialist during daily clinical practice.

## 2. Methods

Anarrative review by means of MEDLINE, The Cochrane Library, and EMBASE was performed by using the following criteria: “apnea, obstructive sleep”, “sleep apnea pathophysiology”, sleep apnea loop gain, sleep apnea arousal threshold”, “sleep apnea muscular impairment”, and “sleep apnea phenotyping”.We focused our selection criteria mainly on original articles, in which we have evaluated the whole information and concepts expressed, to collect the most important and thorough aspects of up-to-date OSA pathophysiology, pathophysiological traits classification, and its identification by means of clinical and polygraphic parameters. As a result,96 out of 3829 abstracts were selected for full-text review, covering a period from 1983 to 2018. Upper airway anatomical collapsibility, neuromuscular tone and function, sleep-wake and ventilatory control instability, and the arousal threshold all interact and contribute to certain pathophysiologic features that characterize different types of OSA.

## 3. Anatomical Collapsibility and Passive Critical Occlusion Pressure (Pcrit)

UA collapse during sleep caused by the specific anatomy of the patient’s characteristics represents one of the most important OSA pathophysiological factors. The coordinated neuromuscular tone of more than 20 muscles controls the pharyngeal function and patency, expanding and stabilizing the human pharynx [14], which has a limited rigid support, being anchored to the skull base and laryngeal cartilage.

Patients with OSA have various reasons for a restricted UA patency [14]. When considering UA collapsibility, it is of pivotal importance to distinguish between skeletal framework abnormalities and soft tissue abnormalities, and this would reflect on static and dynamic components of UA obstruction. Furthermore, with some skeletal framework abnormality, such as retrognathia or nasal skeletal deformity, the impact of mouth breathing and tongue base retraction can both enhance the collapsibility of UA [14]. Quantitative imaging studies have reported significant differences in both craniofacial and soft tissue structures surrounding the UA in OSA patients [15,16]: small bony structures, which consist of low hyoid bone and/or retrognathia of the jaw, lead to a reduction in pharyngeal space [16,17,18].

Certain characteristics of craniofacial abnormalities are likely to be genetically influenced, as they may occur in family members of OSA patients as well, which might explain, at least in part, the different prevalence associated with gender and ethnicity [19,20,21]. Moreover, radiological studies using magnetic resonance (MR) and computed tomography (CT) methodology have found that the most common sites of occlusion in OSA due to bulky soft tissue are the lateral pharyngeal walls at retropalatal level [22,23,24], which can cause narrowing of this region. Deposition of adipose tissue in the lateral pharyngeal walls, soft palate, uvula, and tongue in obese patients is a frequent occurrence in these patients [25,26].

Obesity contributes to UA narrowing during sleep also via other mechanism than fat tissue deposition. It increases the load on the respiratory pump muscles, increases neural respiratory drive, particularly in the supine posture, and causes increased intra-abdominal pressures [27,28,29]. The increased abdominal fat-mass leads to higher intra-thoracic pressures, reduced transpulmonary pressure gradients, and a restrictive lung function, with obese patients breathing at low lung volumes [29]. This causes a reduction of the lower tracheal traction and its stabilizing action on UA [30,31], which leads to more severe desaturation during apnea episodes [32]. The occurrence and severity of OSA is often increased in the supine position during sleep. Indeed, even in non-obese people, the reduction of lung volumes and consequently rising of the diaphragm and the backdrop of the jaw and tongue are considered to be key contributing factors [33,34].

The trauma on the pharyngeal soft tissue caused by snoring and repeated pharyngeal depressions related to the obstructive events leads to chronic inflammation, which can have important consequences on UA function, sensation, and structure. For example, inflammatory-related soft-tissue edema reduces the diameter of the UA and increases pharyngeal collapsibility. These soft-tissue irritations can alter size and compliance of the UA and may cause functional remodeling between contractile and non-contractile elements of the UA structures [35,36,37].

Furthermore, the rostral shift of small quantities of fluids (100–300 mL) in the neck region at night, which return during sleep from the lower limbs due to the change in posture, has been reported as a relevant factor causing and deteriorating OSA [38,39,40].

Nasal obstruction has been proposed as a contributory cause of OSA. Increased nasal resistance requires significant inspiratory efforts to maintain adequate airflow. This causes higher negative inspiratory pressures and increased propensity for pharyngeal collapse. Patients with nasal obstruction frequently breathe through their mouth. This can further promote UA collapse via mandible retraction. Nevertheless, the importance of nasal resistance in determining the severity of OSA seems to have a secondary rather than primary casual role in most cases [21,22,23,24,25,26,41,42,43]. However, increased nasal resistance may be a major casual factor in certain OSA patient groups including in those with spinal cord injury [44,45].

The above factors impact on UA collapsibility, and their effect can be measured with the critical occlusion pressure (Pcrit). The Pcrit is defined as the endo-pharyngeal pressure associated with UA collapse [11,12]. This method has been developed by using sudden reductions of continuous positive airway pressure (CPAP) levels to assess the properties of passive pharyngeal occlusion, to determine the pressure in which the pharyngeal airway occludes [46].

Pcrit measurements are complex and impractical to establish in clinical practice because of the required equipment, time, and expertise needed. The complexity in obtaining and interpreting the pressures and the difficulties in obtaining defined sleep stages during the maneuvre present further challenges. Although this does not negate physiological usefulness of the concept, it substantially limits clinical availability of this method. Moreover, there is little standardization of the diagnostic protocol, limiting comparability between available studies using this method [46].

## 4. Muscular Upper Airway Gain (UAG)

OSA patients frequently have a narrowed UA. However, no obstructive breathing occurs during wakefulness [47]. This highlights the importance of state-dependent (awake—sleep) decreases in neuromuscular tone and compensatory factors [48,49,50].

Muscles responsible for UA patency can be divided into three groups [51]:Muscles that regulate the position of the hyoid bone, mainly the *geniohyoid* and *sternohyoid* muscles.Tongue base muscles, predominantly the *genioglossus* and other pharyngeal constrictors.Muscles of the soft palate, in particular the *tensor palatini* and *elevator palatini* muscles.

Functionally, these muscles can be classified into phasic (muscle activity increased during inspiration compared to a tonic expiratory activity) and tonic activity pattern (muscle activity remains constant throughout the respiratory cycle, without differences between inspiration and expiration). In this context, the *genioglossus* (GG; phasic muscle) and the *tensor palatini* (tonic muscle) are some of the most important and most studied muscles. During wakefulness, the motor nuclei of the GG receive input from the following [52]:Neurons modulating wakefulness/sleep state (active during wakefulness and less active or inactive during sleep).Neurons in the respiratory centers located in the ventro-lateral medulla oblongata (inspiratory phasic premotor activity, usually active 50–100 ms prior to activation of the diaphragm).UA mechanoreceptors triggered by negative endopharyngeal pressure (inspiratory phasic activity).

Patients with OSA have a higher electromyographic (EMG) activity of the GG during wakefulness compared to healthy controls, reflecting a higher phasic activation due to increased negative endo-pharyngeal inspiratory pressures and an increased tonic activity [53,54,55].

GG activity is diminished during sleep onset in OSA patients compared to control subjects [53]. The compensatory impact of high negative pressures is lost, while in parallel, the tonic state-dependent activation is reduced as well. The tensor palatini muscles gradually loses EMG activity of up to 30% compared to wakefulness [53,54,55].

Pharyngeal muscles control the pharyngeal patency and can limit or completely offset negative effects caused by anatomical factors. The effect of neuromuscular compensation can be measured with the active critical occlusion pressure (active Pcrit), a measure obtained during sleep by manipulating the airway pressure slowly [56].

The interaction between anatomic collapsibility and neuromuscular compensation can be summarized by the two following examples that define whether someone will be affected by OSA or not:Patients in whom the neuromuscular compensation offsets the anatomic collapsibility in all positions and in all sleep stages.Patients affected by OSA because neuromuscular compensation does not offset the anatomic collapsibility in all positions and in all sleep stages.Several additional mechanisms may cause a further reduction of the neuromuscular compensation during sleep:Neuromyopathic denervation [57,58,59,60],Muscular anatomical changes [61], andNeuromechanical coupling of UA muscles [35].

UA collapse during obstructive and no-apneic respiratory events can be recorded and detected by three different flow-limiting patterns in a polysomnography. These are consistent in appearance of airflow reduction despite increased inspiratory efforts [62,63].

The Starling resistor pattern: During an obstructive event, an inspiratory effort is characterized by a first phase with increasing flow and effort in a linear way, followed by a second phase during which flow remains stable and independent of any further inspiratory effort. In this polysomnographic pattern, the inspiratory flow limitation persists for the entire inspiratory effort of an obstructive event.

Intra-event negative effort dependence pattern: During an individual inspiratory effort, first, a phase in which flow increases in a linear way is observed, followed by a decreasing flow signal during the later part of the inspiration, with increasing respiratory effort. The level of inspiratory flow limitation is repeated stereotypically during the entire obstructive event.

Inter- and intra-event negative effort dependence pattern [64]: Following an initial phase during which flow increases in a linear way for every inspiratory effort, flow gradually decreases during the second part of the inspiration. As detected during polysomnography, this inspiratory flow limitation progressively deteriorates breath-by-breath during the obstructive event.

The airflow pattern observed in the recording from the nasal cannula is related to different types of neuromuscular compensation to defend UA patency. More recently, the morphology of the flow curve detected by the pressure transducer has been analyzed to identify the site of the UA collapse [65,66,67].

## 5. Loop Gain (LG)

Ventilatory control helps to maintain the homeostasis of blood gases. It is state- and sleep-stage-dependent, with supra-pontine and metabolic control during rapid eye movement (REM) sleep and predominantly metabolic control during non-REM (NREM) sleep stages. The complexity of metabolic ventilatory control during NREM sleep has been simplified and summarized by the engineering model of loop gain (LG), which consists of a control component (chemoreceptor: controller gain), an exchange component (lung: plant gain), and a connection component (circulation: circulatory delay). The responsiveness of LG is a dimensionless measurement; an LG of >1 is related to a sensitive and hyper-responsive system, which can lead to excessive ventilatory responses and destabilization of the ventilation during sleep, resulting in periodic breathing. Whereas a LG <1 is associated with a more stable system and lower ventilatory responses to any respiratory event during sleep, leading to a readjustment of the ventilatory system [68].

Initially, LG was measured by means of a bi-level positive pressure device and more recently by means of UA continuous pressure devices [69,70,71] or sophisticated mathematical analysis of clinical polysomnography PSG [72,73]. None of these methods to measure LG are currently available in clinical routine.

## 6. Arousal Threshold (AT)

The AT is defined as the level of inspiratory effort, measured by the esophageal (Poes) or epiglottic pressure (Pepi), at which obstructive events terminate with an arousal from sleep [74,75]. For a long time, arousals from sleep have been considered to be unavoidable and necessary for ending an obstructive event. However, in more than 25% of obstructive events, arousals [76] may not be observed at the end of the obstructive event [77,78,79]. Alternatively, neural respiratory drive as the force to generate inspiratory pressures have been used to define the AT [79].

During obstructive respiratory events, factors including increased duty cycle and respiratory frequency, as well as neuromuscular mechanisms to increase airflow, are activated via chemical and mechanical feedback loops. If these mechanisms successfully achieve UA patency and sufficient ventilation (sustainable VE), then an arousal from sleep may not be required to support ventilation. The threshold required to achieve UA reopening, sufficient to achieve a sustainable VE, is defined as threshold of effective recruitment (Ter). Essentially, the relationship between the AT and the Ter determines whether there is an arousal at the end of an obstructive event; the arousal occurs when the AT is lower than Ter or when hyperventilation follows UA reopening and stimulates the arousal center [80]. AT and Ter are dependent on sleep stage and other factors, such as age, drugs, alcohol consumption, sleep fragmentation, and sleep deficiency [81].

The termination of an apneic event by arousal leads inevitably to a respiratory response, which can cause ventilatory and pharyngeal muscular instability and promote obstructive events recurrence [71,75,82]. Moreover, a low arousal threshold causes sleep fragmentation and contributes to excessive daytime sleepiness [82].

Available in research sleep laboratories only, the AT is measured during respiratory events or by manipulating continuous pressure devices to induce airflow limitation and arousal [83]. More recently, mathematical analysis of a clinical PSG has been used to estimate the arousal threshold [84]. Neural respiratory drive as an electrical index of ventilation and AT activating the respiratory muscle pump has also been tested in experimental settings [79]. A simple estimate of the arousal threshold from standard PSG parameters which was validated by using Pepi signals during respiratory events has also been proposed and used in follow-up studies [85,86,87].

## 7. Qualitative Phenotyping Model of OSA

Pharyngeal anatomical collapsibility plays a central role in the pathogenesis of OSA, considering that, without any grade of pharyngeal collapsibility, obstructive events may not occur, even if other non-anatomical pathophysiologic factors are significantly altered [62]. The different conditions impacting on the likelihood for OSA development are summarized in the following model Figure 1:

(a) People without any grade of UA collapsibility do not develop OSA, even if other non-anatomical PT are abnormal.

(b1) Patients with severe UA collapsibility develop OSA, independent of any other non-anatomical PT.

(b2) Patients with an intermediate grade of UA collapsibility develop OSA in relation to the pathologic grade of severity of the other non-anatomical PT (AT, LG, and UAG) or in relation with the grade of UA collapsibility severity.

(b3) Patients with low UA collapsibility potentially develop OSA or not, only in the presence of othersignificant non-anatomical PTs.

Lately, the literature in the field has focused on the importance of customized therapy of OSA patients, which could be based on PTs and not exclusively on the anatomical factors, as has been the case so far [69].

Historically, the focus of attention for surgical intervention was the anatomical pathophysiologic factor with pharyngeal collapsibility. The passive Pcrit represents the main measure of anatomical collapsibility [69]. On the other hand, the active Pcrit defines the anatomical collapsibility balanced by neuromuscular factors [56]. Based on the passive Pcrit, it is possible to classify the anatomical collapsibility of the UA in the following way [88]:>2.5 cmH_2_O—“High Collapsibility”+2.5– 2.5 cmH_2_O—“Intermediate Collapsibility”<−2.5 cmH_2_O—“Low Collapsibility”

Recently, it has been proposed that the PALM classification, where “P” stands for Pcrit, “A” for AT, “L” for LG, and “M” for muscle recovery, can classify patients according to for PTs into three different subgroups [63,89]. Although the PALM classification identifies the PTs characteristics in OSA patients, the passive anatomical collapsibility of the UA represents the point of reference [63]. The PALM classification subgroups are as follows:

PALM 1: This group involves about 23% of OSA patients and is characterized by a high anatomical collapsibility (Pcrit higher than +2.5 cmH_2_O). Weight loss, positional therapy, oral appliance (OA), CPAP, and upper airway surgery are the 1st line treatment as these treatments focus on anatomical factors (anatomical treatments).PALM 2: This is the largest subgroup and involves about 57% of patients with OSA; it is characterized by an intermediate collapsibility (Pcrit between +2.5 and −2.5 cmH_2_O). These patients are potential candidates for anatomical treatments (subgroup 2a) or for a combination of anatomical and non-anatomical treatments (subgroup 2b).PALM 3: This subgroup involves approximately 19% of OSA patients with low collapsibility (Pcrit less than −2.5 cmH_2_O), associated with abnormal non-anatomical PT. These patients are potential candidates for non-CPAP treatment options such as weight loss, OA, oxygen, and drugs which target the loop gain or the AT.

There are sparse data on the relationship between UA surgical treatment and Pcrit or muscular UAG [90]. When the relationship between anatomical collapsibility and UA surgery is taken into account, it is possible to identify three likely outcomes (Figure 2a–c):

Surgical treatment of the UA achieves complete resolution of UA collapsibility, and the patient is cured independently of any other PT grade (Figure 2a).

The surgical treatment of the UA significantly improves pharyngeal collapsibility up to a low grade of collapsibility. In this model, any complete resolution of apneic events or the persistence of OSA are caused by confounding PT (Figure 2b).

Surgical treatment of the UA does not achieve any significant improvement of pharyngeal collapsibility, which remains of high grade, and OSA does not improve (Figure 2c).

The above highlights the importance that the otorhinolaryngologist should be aware of PT characteristics and, particularly, of upper airway anatomical collapsibility when discussing treatment options with patients.

Recently, novel methods of PT analysis have been introduced using clinical PSG and mathematical models [72,85,91]. So far, laboratory procedures to determine PT are mostly limited to the research setting and are unavailable in routine clinical practice. Given the complexity of OSA pathogenesis, it is quite possible that they will not be implemented into routine clinical practice in the near future, despite the clear potential to improve treatment outcomes. Data concerning the qualitative PT measurement by means of clinical PSG and clinical data have been reported [73]. Such an approach would allow qualitative phenotyping with widely accessible methodology in a routine clinical scenario. Table 1 reports the model based on PSG variabilities required for qualitative phenotypization, deduced from clinical examinations and procedures [73].

The following data regarding UA anatomical collapsibility assessment are extrapolated from the published literature providing useful information for qualitative phenotypization:

### 7.1. Polysomnographic Pattern 

• A polysomnographic predominant apneic pattern (AHI characterized by ≥90% obstructive apneic events) is related to a high UA collapsibility (Pcrit > 2.5 cmH_2_O, PALM 1) (Figure 3) [92,93]

• Upper airway resistance syndrome (UARS) is identified by a polysomnographic picture with normal AHI but is associated with a high prevalence of respiratory-effort-related arousals (RERAs); it is also associated with mild UA collapsibility (Pcrit less than −2.5 cmH_2_O, PALM 3) (Figure 4) [64].

### 7.2. Therapeutic level of CPAP

A therapeutic level of CPAP of less than 8 cmH_2_O differentiates OSA patients with a Pcrit less than -2.5 cmH_2_O (mild collapsibility, PALM 3) from patients with Pcrit higher than –2.5 cmH_2_O (moderate and high collapsibility, PALM 2 and 1) [64].

### 7.3. Severity of OSA measured by the apnea hypopnea index (AHI)

According to data published by Eckert et al., it may be possible to get some insight into PT based on Pcrit and AHI [64]:(A)Patients with a Pcrit> 2.5 cmH_2_O (high collapsibility, PALM 1) typically have a high AHI > 40/h.(B)Most patients with a Pcrit less than −2.5 cmH_2_O (low collapsibility, PALM 3) have an AHI < 40/h.(C)Patients with a Pcrit between –2.5 and +2.5 cmH_2_O (moderate collapsibility, PALM 2) can stratify at each level of AHI severity.

However, the AHI in and of itself is not very helpful in differentiating different PTs. Instead, combined approaches such as an AHI < 40/h and a therapeutic level of CPAP ≤8 cmH_2_O allow for a more accurate subclassification of OSA patients than the use of single parameters (Table 2).

In addition to the anatomical collapsibility classification, it is possible to achieve a qualitative assessment of the other PTs (LG and AT) with a similar strategy. Table 3 shows the most important parameters and PSG patterns for LG and AT qualitative phenotyping [73]. In summary, combining the results of the Pcrit, LG, and AT scoring (Table 4) may help to obtain a meaningful qualitative phenotypization in OSA patients.

A careful qualitative analysis of PTs using clinically available data and exams provides useful information for planning therapeutic strategies, which can improve pre-operative surgical evaluation and the quality of information available to the patients. Using this approach, it is possible to identify at least two main clinical scenarios that are clinically useful for the sleep specialist:(1)High collapsibility of the UA: In these cases, the presence of impaired LG and AT represents a negative predictive factor for any anatomical surgical outcome, as compared to patients in which the UA collapse represents the only PT impaired.(2)Low UA collapsibility: Patients with low anatomic collapsibility may benefit from targeted non-CPAP therapy (weight loss, mandibular advancement device (MAD), sleep stabilizing drugs, and oxygen). Any surgical approach in these patients should be reserved as a second-line therapeutic option. This subset of patients can reliably be identified.

## 8. Conclusions

While additional simple, accurate quantitative estimates are required to translate phenotyping concepts into widespread clinical practice, it is possible to propose a qualitative practical phenotyping model of OSA by analyzing PTs based on a limited number of available clinical and polysomnographic parameters, which analyze the UA anatomical collapsibility, the parameter of greatest interest for any sleep specialist. This information is a useful addition to the pre-operative surgical assessment to guide therapeutic recommendations and inform patients about potential outcomes.

The current review was designed and conducted in compliance with the principles of Good Clinical Practice regulations and the Helsinki declaration.

## Figures and Tables

**Figure 1 ijerph-17-02058-f001:**
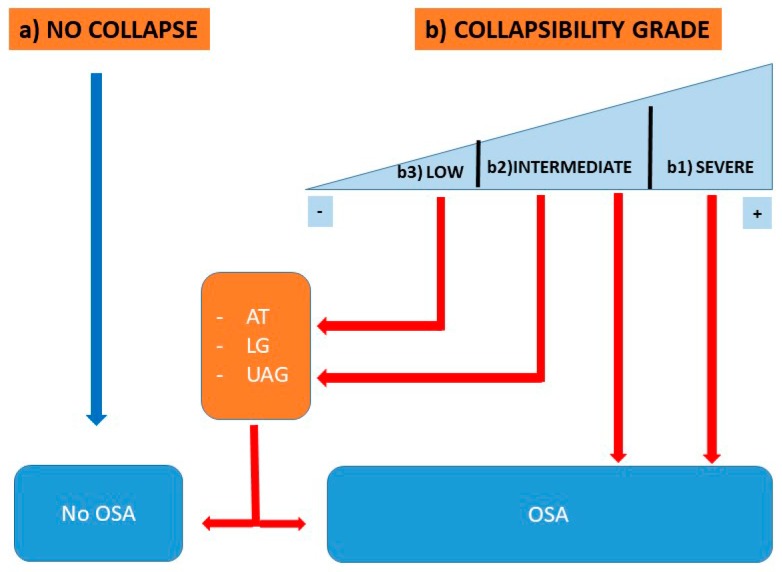
Relationship between UA Collapsibility and other PTs. Patient (**a**) has no UA collapse and does not develop OSA, even if the other PTs are impaired. Patient (**b1**) has a severe UA collapsibility and develops OSA even if the other PTs are normal. Patient (**b2**) has an intermediate level of UA collapsibility. Patient (**b3**) has a mild degree of UA collapsibility and develops OSA only if the other PTs are significantly impaired. Legend: AT: arousal threshold; LG: loop gain; UAG: upper airway gain; OSA: obstructive sleep apnea.

**Figure 2 ijerph-17-02058-f002:**
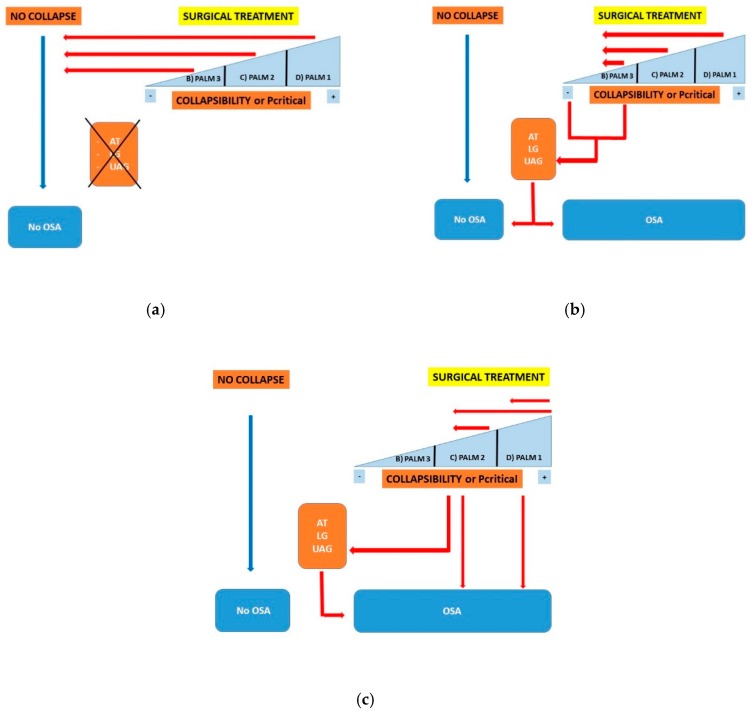
(**a**) Surgical treatment of the upper airways (UA) achieves complete resolution of UA collapsibility, and the patient is cured independent of any other pathophysiological trait (PT) grade. (**b**) Surgical treatment of the UA significantly improves pharyngeal collapsibility up to a low grade of collapsibility. In this model, any complete resolution of apneic events or the persistence of OSA are caused by confounding PT. (**c**) Surgical treatment of the UA does not achieve any significant improvement of pharyngeal collapsibility, which remains of high grade, and obstructive sleep apnea (OSA) does not improve. Legend: PALM: Pcrit, AT: arousal threshold; LG: loop gain; UAG: upper airway gain; OSA: obstructive sleep apnea, PT: pathophysiological trait.

**Figure 3 ijerph-17-02058-f003:**
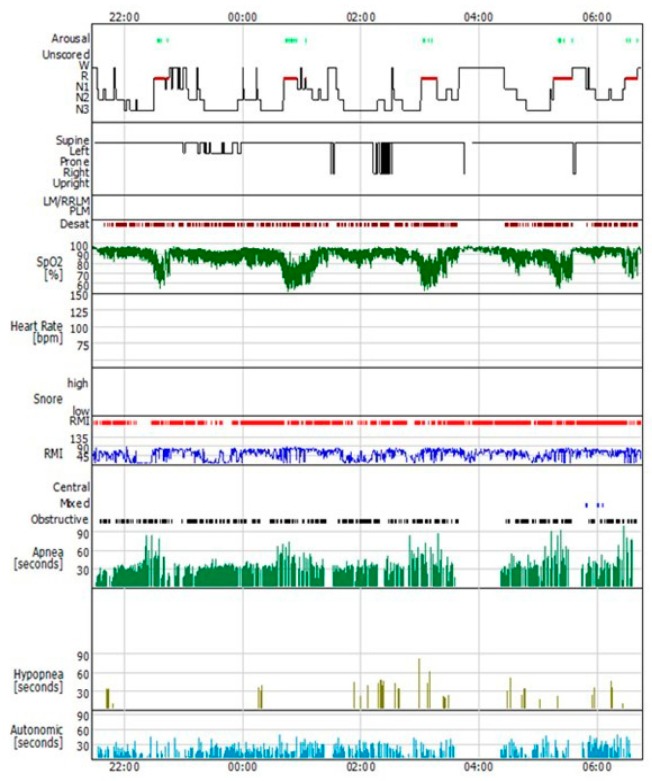
Predominantly obstructive apneic events (OSA). OSA with more than 90% of the respiratory events are obstructive apneas, characterized by a high UA collapsibility. Legend: OSA: obstructive sleep apnea; UA: upper airways.

**Figure 4 ijerph-17-02058-f004:**
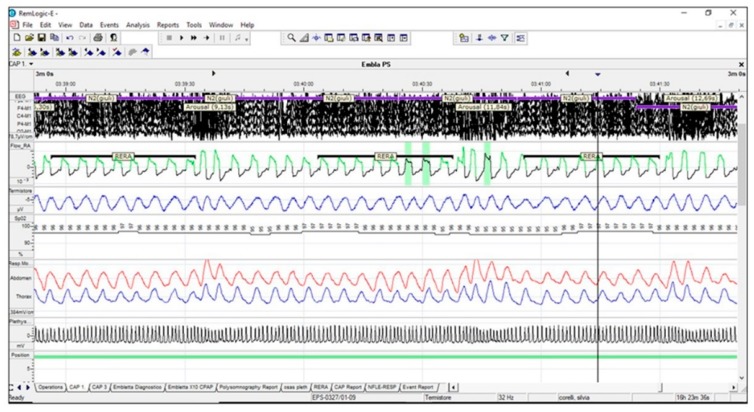
UARS is characterized by low UA collapsibility. Airflow is slightly limited with ever-increasing respiratory effort, until arousal from sleep occurs. No arterial desaturations are observed with RERAs, but these respiratory events lead to sleep fragmentation. Legend: UARS: upper airways resistance syndrome; UA: upper airways; RERA: respiratory-effort-related arousals.

**Table 1 ijerph-17-02058-t001:** List of patterns and polysomnographic (PSG) variables allowing a qualitative assessment of pathophysiological traits.

**(I) ANATOMIC COLLAPSIBILITY**	**(II) AROUSAL THRESHOLD**
PALM 1 or 2 (1) Obstructive Apnea pattern (High Pcrit) (2) Severe AHI (High Pcrit) PALM 3 (1) UARS pattern (Low Pcrit) (2) CPAP value ≤ 8cm H_2_O (Low Pcrit)	LOW AT (1) At least 2 out of 3 PSG VARIABLES: AHI <3O Hypopnea/Apnea > 58.3% Nadir > 82.5% (2) UARS pattern HIGH AT Duration and severity of desaturations
**(III) VENTILATORY INSTABILITY**	**(IV) MUSCULAR RECOVERY**
(1) Coexistence of OSA and CSR (2)High proportion of central/mixed event (3) N-REM predominant patterns	(1) Starling resistor pattern (2)Intra-breath negative dependence pattern (3) Intra-event negative dependence pattern

**Legend:** PALM: P stands for critical pressure, A for arousal threshold, L for loop gain, and M for muscle recovery; AHI: apnea hypopnea index; UARS: upper airway resistance syndrome; CPAP: continuous positive airway pressure; N-REM. Non rapid eye movement.

**Table 2 ijerph-17-02058-t002:** Relation between collapsibility, AHI, CPAP value, and PALM classification.

Collapsibility.	AHI	CPAP Value	PALM Classification
less than −2.5 cmH_2_O	<40	≤8 cmH_2_O	PALM 3
between −2.5 and +2.5 cmH_2_O	<40	>8 cmH_2_O	PALM 2
more than −2.5 cmH_2_O	>40	>8 cmH_2_O	PALM 2 or 1

Caption: By using the AHI (cut-off 40 events/h), in parallel with the effective pressure value (CPAP ≤ 8 cm), it is possible to characterize the collapsibility of relatively high percentage of OSA patients. Legend: AHI: Apnea Hypopnea Index; CPAP: continuous positive airway pressure; OSA: obstructive sleep apnea. PALM: stands for critical pressure, A for arousal threshold, L for loop gain, and M for muscle recovery.

**Table 3 ijerph-17-02058-t003:** Polysomnographic (PSG) patterns consistent with ventilatory instability and impaired AT [73].

Ventilatory Instability	Arousal Threshold (AT)
(1) Coexistence of OSA and CSR (2) High proportion of central/mixed respiratory events (3) NREM dominant pattern (CAP dominant OSA)	Low AT (1) At least 2/3 variables (AHI<30, H/A> 58.3%, SaO2 Nadir >82.5%)High AT (2) Long event duration and severe desaturations (e.g., OHS)

Legend: AT: arousal threshold; CSR: Cheyne–Stokes Respiration; CAP: cyclic alternating pattern; OHS: obesity hypoventilation syndrome.

**Table 4 ijerph-17-02058-t004:** The upper airway (UA) collapsibility, arousal threshold, and loop gain scoring.

UA COLLAPSIBILITY	LOW AROUSAL THRESHOLD	HIGH LOOP GAIN
high collapsibility	yes/no	yes/no
intermediate collapsibility	yes/no	yes/no
low collapsibility	yes/no	yes/no

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
