# Peer review of "Qualitative Phenotyping of Obstructive Sleep Apnea and Its Clinical Usefulness for the Sleep Specialist"

_ijerph, 2020, doi:10.3390/ijerph17062058_

Round 1
Reviewer 1 Report
This is a very interesting and well-thought-out paper.
My concerns are mostly in the stylistic realm, where what seem to be incomplete sentences are present (i.e. lines 81-83), grammatical errors (line 60 should not have an apostrophe after "selection"), or other simple mistakes (line 47-48 "Excessive sleepiness" seems to be out of place.
Other concerns include line 65 - why were only 96 abstracts chosen? no clear information about that is given
line 114 - "UA collapse of the UA" seems to be a mistake
line 217 - "neuromuscular" should not be in quotes
line 229 - this line should be cited
The figures and the description of PALM was very nice
Overall this could be a great paper but just needs some more fine tuning before I can recommend for publication
Author Response
All the authors thank the reviewer for the overall positive judgment about the paper.
Here our answers to your comments and suggestions:
Line 47-48: all the sentence was not complete in the version for the reviewers, but it was in the file sent to editor. We have add again all the sentence.
Line 81-83, now 84-86: we have modified the sentence, improving the overall meaning.
line 114, now 117: corrected
line 217, now 220: corrected
line 65, now 68: we have selected 96/3829 abstracts because it is a narrative review, not a systematic review. We have clarified this concept the new version of our manuscript, also following the same suggestion of another review.
We have cancelled the "systematic" word form the text in the method section. See line 60
line 229, now 232: a quote has been added
Reviewer 2 Report
Major comments
The title is about qualitative phenotype and their clinical usefulness. However, the introduction is feeble because they only cover epidemiology and anatomical/physiological aspects. The introduction is very short and not allow to understand the general idea of the manuscript
Do systematic reviews follow some guidelines? (e.g. PRISMA)
The “methods” probably is one the most critical parts of a systematic review, because allow knowing the rigour with which it was made. But in this case, the information is insufficient. It is very short, and essential information is missing.
Was there any analysis of the bias of the selected articles?
In results, there is no information on how these 96 definitive articles were reached.
Finally, this text is not a systematic review. Instead, it seems like a narrative review, so the authors must make profound changes to qualify as a systematic review.
Author Response
All the Authors thank the interesting comments and suggestions sent by the reviewer.
1) regarding the introduction: 4 lines about the symptoms and potential complications of OSA were missed from the original version sent to the editor. Now we have added it again.
2) We have specified in the revised version of our manuscript that our paper is a narrative review and not systematic. We canceled the term systematic from the Materials and Methods because we acknowledge that it could be a misunderstanding.
3) We apology for the misunderstanding, but we remark that our paper is a narrative review.
Reviewer 3 Report
Greetings Authors,
The review article is clearly presented. However, some minor suggestions can increase the importance of this paper.
- Various pathophysiological traits that are unique to the obstructive sleep apnea population are discussed comprehensively.
- Introduction part: Some parts of the introduction with references are missing (2-12).
- In division 5. The clinical phenotype of OSA based on age, sex, and ethnicity are only briefly discussed. As there is a lot of difference in the Craniofacial features among the Asian, African American, and Caucasian population, it will be worth to elaborate this in the section.
- Rather than reviewing the clinically defined phenotypes like excessive sleepiness (EDS) or Genetic and molecular markers, authors focused on the PTs, which will be valuable in analyzing the outcome of the surgical treatment.
Author Response
All the Authors thank the interesting comments and suggestions sent by the reviewer.
2) 4 lines in the introduction session has been missed from the original file sent to the editor, with the proper references. We have added it again.
3) the Authors focused their attention only on the general concepts about craniofacial abnormalities and OSA because it could be long-winded. We acknowledge it would be interesting and a matter for further studies.
Round 2
Reviewer 2 Report
I think the authors improved the quality of their manuscript; however, it does not meet the conditions for publication. In the current format, it becomes tedious to read. In some parts, loss the structure, for example, in the discussion. Major comments The keywords must be words used in the abstract. CPAP and PALM classification, are not used For why reason the authors choose 1983 as the year of start? Do the authors use specific words for the searching? Please, add in the methods section. Apparently, the titles number 5 (line 72), 6 (line 133), 7 (line 197), 8 (line 212) are parts of results. I suggest that they do not use numbers. The authors can use 4.1, 4.2, 4.3, and the discussion must be use number 5. Please, write the name of muscles in italic. The authors need to improve figure 2 (a,b, and c). Please merge into one. Ideally, all three in the same line to notice the differences. As they are separated, it is difficult to understand the differences. The discussion section is commonly used to compare the literature and discuss the agreements and differences. However, the authors incorporate information that seem results, for example, the PALM classification, the polysomnographic pattern, the therapeutic level of CPAP. I suggest that this narrative review not be written as an original article (Introduction, methods, results, discussion, and conclusion), but rather as a continuous text with titles and subtitles, as is traditionally done in narrative reviews. Minor comments Abstract Line 28: Please delete the word “systematic.” Line 30: Please, delete the words “ peer-reviewed.” It is not necessary to report this, because we expect that all reviews use peer-reviewed evidence. Line 36: Please, delete the words “peer-reviewed.” Methods Line 60: Please replace “specific” by “narrative.” Results: Line 67: Replace the number “4” by “3” Line 81-83: This sentence needs to be referenced in the bibliography. Line 105-107: This sentence needs to be referenced Line 157-159: This sentence needs to be referenced Discussion Line 261: Please, delete “peer-reviewed” Line 274: Use the number in order (63, 91)Author Response
All the Authors thank the reviewer for the new comments and suggestions.
We introduced several changes to the text, trying to match the reviewer's expectations.
- Reviewer: I think the authors improved the quality of their manuscript; however, it does not meet the conditions for publication. In the current format, it becomes tedious to read.
Response: all the Authors are sorry that the reviewer considers the current format tedious to read.
We hope that with the further changes requested, the text will be more in line with the reviewer's expectations.
- Reviewer. Major comments The keywords must be words used in the abstract. CPAP and PALM classification, are not used
Response: PALM classification term has been canceled from the keywords line. CPAP has been added to the abstract text.
- Reviewer. For why reason the authors choose 1983 as the year of start? Do the authors use specific words for the searching? Please, add in the methods section.
Response: 1983 as the year of start is a result, not our first decision. In the current text this concept has been specified better. In the Methods section, we introduced the terms used for the literature search.
- Reviewer. Apparently, the titles number 5 (line 72), 6 (line 133), 7 (line 197), 8 (line 212) are parts of results. I suggest that they do not use numbers. The authors can use 4.1, 4.2, 4.3, and the discussion must be use number 5.
Response: Considering that all the manuscript has been modified as narrative review, leaving only the methods paragraph, all the other paragraphs such as Anatomical Collapsibility and Passive Critical Occlusion Pressure (Pcrit), Muscular Upper Airway Gain (UAG) etc maintain titles numbers.
- reviewer. Please, write the name of muscles in italic. The authors need to improve figure 2 (a,b, and c). Please merge into one. Ideally, all three in the same line to notice the differences. As they are separated, it is difficult to understand the differences.
Response: the muscles' name is in italic in the current text.
All the Authors disagree about reviewer comments regarding figure 2a,b,c. We think that put all three figures together would be of great misunderstanding for the readers. Reading the text and the figure description, the meaning of the single figure is clear and useful.
- The discussion section is commonly used to compare the literature and discuss the agreements and differences. However, the authors incorporate information that seems results, for example, the PALM classification, the polysomnographic pattern, the therapeutic level of CPAP. I suggest that this narrative review not be written as an original article (Introduction, methods, results, discussion, and conclusion), but rather as a continuous text with titles and subtitles, as is traditionally done in narrative reviews.
Response: all the text has been revised as a narrative review.
- Minor comments Abstract Line 28: Please delete the word “systematic.” Line 30: Please, delete the words “peer-reviewed.” It is not necessary to report this, because we expect that all reviews use peer-reviewed evidence. Line 36: Please, delete the words “peer-reviewed.” Methods Line 60: Please replace “specific” by “narrative.” Results: Line 67: Replace the number “4” by “3” Line 81-83: This sentence needs to be referenced in the bibliography. Line 105-107: This sentence needs to be referenced Line 157-159: This sentence needs to be referenced Discussion Line 261: Please, delete “peer-reviewed” Line 274: Use the number in order (63, 91)
Response: all the corrections suggested by the minor comments have been introduced in the text.
Lines 105-107: the references for the concepts expressed in these lines are reported at the end of the paragraph as numbers 35, 36,37.
Round 3
Reviewer 2 Report
I consider the article to be substantially improved and ready for publication.
Minor comments:
Replace in the filiation number 1 and 7 “Pulmorany” by “Pulmonary”